# Approaching diamond's theoretical elasticity and strength limits

Anmin Nie[1,6], Yeqiang Bu[2,6], Penghui Li[1,6], Yizhi Zhang[2,3], Tianye Jin[4], Jiabin Liu[2], Zhang Su[1], Yanbin Wang[5], Julong He[1], Zhongyuan Liu[1], Hongtao Wang[2,3]*, Yongjun Tian[1]* & Wei Yang[2,3]

Diamond is the hardest natural material, but its practical strength is low and its elastic deformability extremely limited. While recent experiments have demonstrated that diamond nanoneedles can sustain exceptionally large elastic tensile strains with high tensile strengths, the size- and orientation-dependence of these properties remains unknown. Here we report maximum achievable tensile strain and strength of diamond nanoneedles with various diameters, oriented in <100>, <110> and <111> -directions, using in situ transmission electron microscopy. We show that reversible elastic deformation depends both on nanoneedle diameter and orientation. <100> -oriented nanoneedles with a diameter of 60 nm exhibit highest elastic tensile strain (13.4%) and tensile strength (125 GPa). These values are comparable with the theoretical elasticity and Griffith strength limits of diamond, respectively. Our experimental data, together with first principles simulations, indicate that maximum achievable elastic strain and strength are primarily determined by surface conditions of the nanoneedles.

[1] Center for High Pressure Science, State Key Laboratory of Metastable Materials Science and Technology, Yanshan University, 066004 Qinhuangdao, China. [2] Center for X-mechanics, Zhejiang University, 310027 Hangzhou, China. [3] Institute of Applied Mechanics, Zhejiang University, 310027 Hangzhou, China. [4] Center for Precision Engineering, Harbin Institute of Technology, 150001 Harbin, China. [5] Center for Advanced Radiation Sources, University of Chicago, Chicago, IL 60439, USA. [6]These authors contributed equally: Anmin Nie, Yeqiang Bu, Penghui Li. *email: htw@zju.edu.cn; fhcl@ysu.edu.cn

The ideal strength of a crystalline solid is defined as the maximum stress a perfect crystal lattice can withstand at 0 K[1,2]. However, virtually no practical bulk materials survive under the ideal maximum stress because of the presence of internal defects and surface flaws. Based on experimental observations, Griffith[3] proposed a more useful theoretical tensile strength of ~$E/9$, where $E$ is the Young's modulus of the solid. With further development[4–6], the theoretical tensile strength is considered of the order of $E/10$. Two facts regarding defects guide the direction in designing ultra-strong materials toward reducing material dimension to nano- or even atomic-scales:[7,8] (1) probability of finding a defect decreases with material volume, and (2) maximum size of defect is limited by the overall dimension of the material. By reducing material dimensions, internal defects and surface flaws are significantly reduced; hence material failure is primarily controlled by the intrinsic limits of atomic bonds, making it possible to achieve the theoretical strength of the material[9–11].

Near theoretical strengths have been achieved for silicon nanowires (20 GPa)[12], carbon nanotubes (>100 GPa)[13], and Graphene (100–130 GPa)[8,11] in nanomechanical tests. Particularly, uniaxial tensile strength of graphene has been shown to reach the theoretical limits of $E/9$, the highest tensile strength experimentally achieved to date[14]. Because of the extremely high bulk modulus and hardness, diamond has historically been considered as the strongest bulk material[15]. However, it is still challenging to directly measure the tensile strength of diamond due to its poor deformability and relatively high brittleness[16,17]. Mechanical properties and fracture behavior of diamond are mostly tested by indentation[18,19] or compression in the diamond-anvil cell[20]. Based on Hertzian indentation, the tensile strength of diamond is measured to be 20 GPa[21], which is far below the ideal strength of ~225 GPa calculated by first principles[22,23] and Griffith theoretical strength of ~122 GPa. The low tensile strength of bulk diamond is primarily attributed to inelastic relaxation induced by the movements of defects and the premature failure caused by the propagation of microcracks[21,24].

Notably, diamond nanoneedles with sub-micrometer diameters can be reversibly deformed with local tensile strains up to ~8.9% by in situ bending inside a scanning electron microscope[25], corresponding to an estimated maximum tensile stress of ~98 GPa. The scarcity of internal defects coupled with the smoothness of the surface is considered the key to reaching this ultrahigh tensile strength. Although this work provides a strategy for elastic strain engineering, the lack of understanding of size, orientation and surface-defect dependence of elastic deformation prevents quantified specifications of diamond properties for electronic and optical applications. Here we tackle this challenge by measuring deformation of high-surface-quality diamond nanoneedles with various diameters and orientations.

We fabricate high-quality single-crystalline diamond nanoneedles using focused ion beam (FIB) milling coupled with argon plasma thinning. Either <100>, or <110>, or <111> directions are along the axes of the nanoneedles. State-of-the-art nanomechanical bending experiments are conducted in situ inside a transmission electron microscope (TEM). Experimental observations are complemented with detailed computational simulations by the finite element method (FEM) and first principles calculations to determine local strain and stress prior to failure and to understand atomistic mechanisms of fracture.

## Results

### Fabrication and characterization of diamond nanoneedles.
Figure 1a is a high-angle annular dark-field scanning TEM (HAADF-STEM) image of a typical diamond nanoneedle, with a

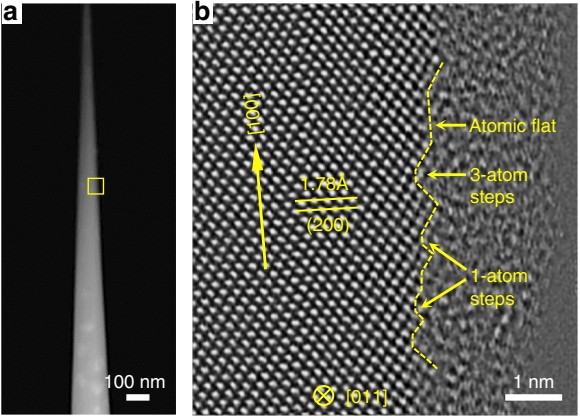

**Fig. 1** Characterization of a <100> diamond nanoneedle. **a** A low-magnitude high-angle annular dark-field scanning TEM (HAADF-STEM) image of a diamond nanoneedle after fabrication. The needle axis is parallel to [100]. **b** Atomically resolved annular bright-field scanning TEM (ABF-STEM) image of the free surface as marked by the yellow box in **a**.

tip around 20 nm in diameter. The bright patches (near the bottom of Fig. 1a) are due to the {001} platelet defects in type-Ia diamond[26]. These defects are mainly located in regions with diameters larger than 100 nm (Supplementary Fig. 1a). Figure 1b is an atomic scale bright-field STEM (BF-STEM) image (corresponding to the box in Fig. 1a), viewed along the [011] zone axis with the [100] direction nearly vertical. Internal defects can hardly be found in parts of the nanoneedle with diameters <100 nm, and the surface of the nanoneedle is featured by atomically flat facets separated by 1–3 atom steps, which are a common feature of the surface of all the nanoneedles we prepared (Supplementary Fig. 1c, d). The residual amorphous layer on the surface due to FIB milling is minimized to be ~2 nm thick after argon plasma cleaning, effectively minimizing the influence of amorphous carbon on the mechanical properties of the diamond needles. The low defect concentration and smooth surface of the diamond nanoneedles enhance the possibility of achieving ultrahigh strength and large fracture strain that has ever been reached.

**Elastic deformation of diamond nanoneedles.** During the tests, a diamond nanoneedle was driven to gently touch the diamond indenter and gradually bent under step-by-step displacement loading (Supplementary Movie 1). All diamond nanoneedles tested exhibited ultrahigh elastic deformability. Figure 2a, b, and d are typical TEM images of a [100] nanoneedle prior, during, and after the bending test, respectively, indicating fully reversible elastic deformation. We calculated the strain distribution of the nanoneedle as shown in Fig. 2b by FEM (Fig. 2c). A tensile strain of 10.1% was inferred, without fracture (Fig. 2c). Such a state of ultrahigh elastic deformation was repeatedly realized in other <100> nanoneedles with similar diameters (Supplementary Movie 2, 3). The fully elastic behavior is due to the relatively low uniaxial tensile or compressive stress during bending at nm scales, so that dislocations cannot nucleate and plasticity does not initiate[17,27]. We investigated lattice distortion in the diamond nanoneedle during bending by selected area electron diffraction (SAED). The initial $d$-spacing of (200) lattice planes of undeformed nanoneedle is measured to be 1.78 Å (Fig. 2e). Figure 2f shows the SEAD pattern of the bent nanoneedle in the circled area in Fig. 2b. This SAED pattern is rotated by 32° relative to that of Fig. 2e, and the (200) spot elongates radially, suggesting varying d-spacing distribution from 1.7 to 1.94 Å. This latter value implies a (200) lattice expansion of 9%, in broad agreement

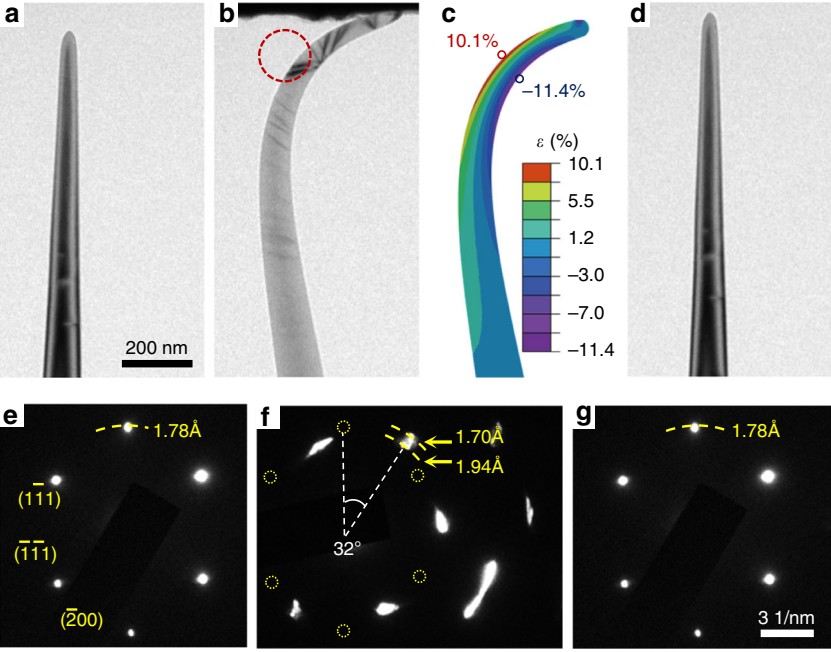

**Fig. 2** An example of reversible bending tests and associated lattice expansion determined by electron diffraction. **a** A TEM image of a <100> diamond nanoneedle with a tip diameter of ~50 nm. **b** The same nanoneedle during compression. **c** Finite element methods (FEM) simulation reproducing the shape in **b** revealing the maximum tensile strain (10.1%) located at the red circle. **d** The same nanoneedle, after unloading. The nanoneedle returns to its original shape. **e–g** Selected area electron diffraction (SEAD) patterns taken during the bending test at strain states of **a**, **b**, **d**, respectively. **f** was taken from the highly curved region (the red circled) in **b**.

with that calculated by FEM. Upon retracting the load, the nanoneedle unbent, and the corresponding SAED pattern instantaneously recovered to the initial state (Fig. 2g), indicating fully reversible elastic deformation. Such a significant and reversible structural response to large deformation renders a considerable range of property tuning opportunities for diamond nanoneedles.

**Fracture behaviors of diamond nanoneedles.** Figure 3 shows a typical breaking sequence of a <100> nanoneedle, as recorded in Supplementary Movie 4. After each test, the long, tapered nanoneedle would break near the tip, leaving a new tip with a larger diameter, and the remainer of the needle returned to its unstrained state. This allowed us to investigate size (diameter) effect on tensile strain and strength using the same nanoneedle, without complicated effects due to switching the test piece. The snapshots in Fig. 3a1-a4 capture the maximum deformation of the nanoneedle immediately before each breaking point. With the nanoneedle breaking at diameters of 60, 95, 115, and 150 nm, respectively, the maximum achievable elastic tensile strains were 13.4, 9.0, 11.1, and 6.5% (Fig. 3b1-b4), and the corresponding maximum local tensile stresses were 125, 88, 105, and 65 GPa, respectively, according to FEM simulations. Besides the tensile strain on the convex side of the nanoneedles, there must be a compressive strain on the concave side for mechanical equilibrium. FEM calculations show that the absolute values of the compressive strain on the concave side are slightly larger than those of the corresponding tensile strain at the convex side (Fig. 2c, Fig. 3b, Fig. 4c, Fig. 4d, Supplementary Fig. 3b and 4b). The tensile strength of a material is usually much lower than the compressive strength because compressive loading tends to suppress microcracks. It has been experimentally demonstrated that compressive strains of −19 and −16% can be achieved in the <100> and <111> -oriented diamond pillars with a diameter of 200 nm in uniaxial compression, respectively[17]. Here, the

compressive strains at the concave side of diamond nanoneedles are still far below the limits of the nanoneedles. Therefore, fracture tends to initiate from the convex side and the tensile limits of the nanoneedles can be well expressed by the bending experiments[25]. TEM observations show that the fracture surfaces of the nanoneedle (Fig. 3c1-c4) consist of either atomically flat {111} planes or {111}-facets (Supplementary Fig. 2), indicating that {111} plane cleavage is the dominant failure mode.

The above results reveal a strong size dependence on elastic deformation of the <100> diamond nanoneedles. We also conducted bending tests on <110> - and <111> nanoneedles. Detailed bending and fracture processes are given in Supplementary Movie 5, 6, respectively. The original state and maximum deformation of the nanoneedles immediately prior to breaking are shown in Fig. 4a, b. Deformation can be precisely simulated by FEM (Fig. 4c, d). Typical sequential breaking processes of <110> - and <111> nanoneedles are shown in Supplementary Figs. 3, 4, respectively. Maximum achievable elastic strains are 9.6 and 9.4% for <110> - and <111> nanoneedles, respectively, after repeated experiments.

We summarize all the experimental data in Fig. 4e. The scatter in the data reflects directly the statistical nature of brittle fracture in diamond. Fracture tends to initiate from defects, which may be randomly distributed[1]. Nonetheless, maximum achievable elastic strain of diamond nanoneedles clearly depends on size and orientation. <100> nanoneedles consistently exhibit higher fracture tensile strains than those of <110> and <111> nanoneedles at the same diameter. The maximum tensile strain of 13.4 % in our tests is ~50% larger than the 8.9% recently reported[25] and is the largest that has ever been experimentally achieved so far.

**Discussion**

We calculated stress-strain relations under uniaxial tensile for single-crystal diamond along [100], [110], and [111] by first

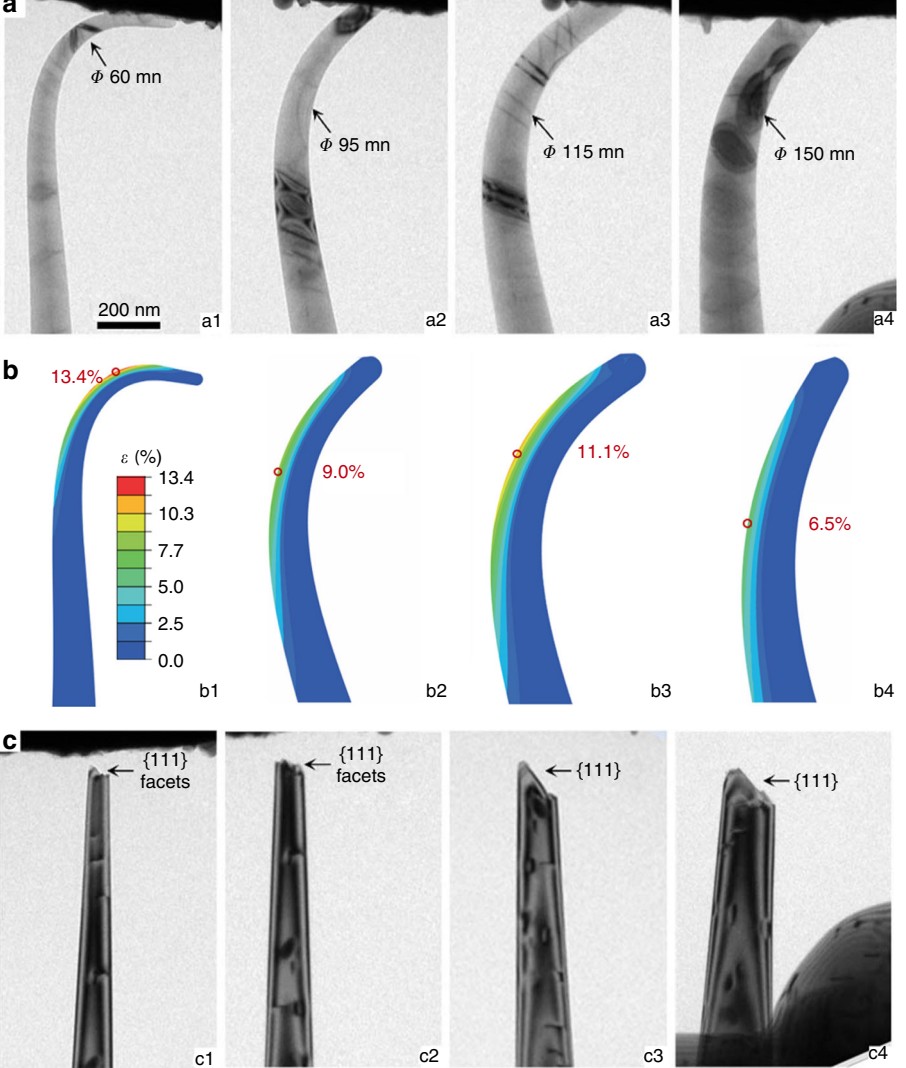

**Fig. 3** A sequence of breaking tests on a single <100> - diamond nanoneedle. Arrows indicate locations of fractures in subsequent breaking tests. **a** Snapshots (a1–a4) capturing the maximum deformation immediately before the fracture during sequentially breaking the diamond nanoneedle for its high aspect ratio geometry. For more details, see Supplementary Movie 4. **b** FEM simulations reproducing the critical needle geometry immediately prior to breaking in **a** and with maximum principle strain distribution in the nanoneedle. **c** TEM images of the needle after the corresponding breaking tests, revealing that all fracture surfaces consist of {111} facets.

principles calculations (Fig. 5a), to compare with our experiments. The stress maxima in stress-strain curves are the ideal strengths, which are 225, 126, and 92 GPa under uniaxial tension parallel to [100], [110], and [111], respectively, consistent with previous theoretical results[15,22,23]. Following these ideal stress-strain curves, our measured maximum fracture strains of 13.4, 9.6, and 9.4% for <100>, <110>, and <111> nanoneedles, correspond to tensile fracture strengths calculated from the stress-strain curves are 125, 84, and 82 GPa. These values are in excellent agreement with our FEM simulations. In case of <100> nanoneedles, Griffith theoretical strength (122 GPa) has been reached.

When the ideal strengths along [100] and [110] are projected to [111], the resultant values of 109 and 95 GPa, respectively, are in good agreement with the ideal strength of 92 GPa along [111] (The detailed description of the projection is shown in Supplementary Fig. 5). This supports the observation that all fractures initiate from {111}. To further understand physical origins of the lower fracture strengths along <110> and <111>, we examine variations of Young's moduli along [100], [110], and [111] as a

function of strain in Fig. 5b. According to Born and Huang's criteria[28], mechanical instability occurs at the point where any elastic modulus approaches zero. Such instability is inherently correlated with changes of atomic configurations at large strains. Along any of the three uniaxial tension directions, C–C bonds elongate monotonically with increasing strain (Fig. 5b). Details of the C–C bond response, however, vary with orientation. An abrupt change of bond length occurs at 1.84 Å when strained along [111], followed by bond breaking. Mechanical instability occurs before this breaking point. The critical bond length corresponding to zero elastic modulus is around 1.78 Å, where an infinitesimal load increase will lead to fracture. As the set of excessively elongated bonds is along the [111] direction, the mode of instability appears as the (111) plane cleavage. For diamond stretched along [110] and [100], progressively larger strains are required to reach zero Young's moduli (top panel of Fig. 5b). Interestingly, for all three orientations, the critical bond lengths are within a narrow range of 1.75–1.78 Å. This implies that failure mode is controlled by cleavage in {111} regardless of uniaxial loading directions and is consistent with the observation that all

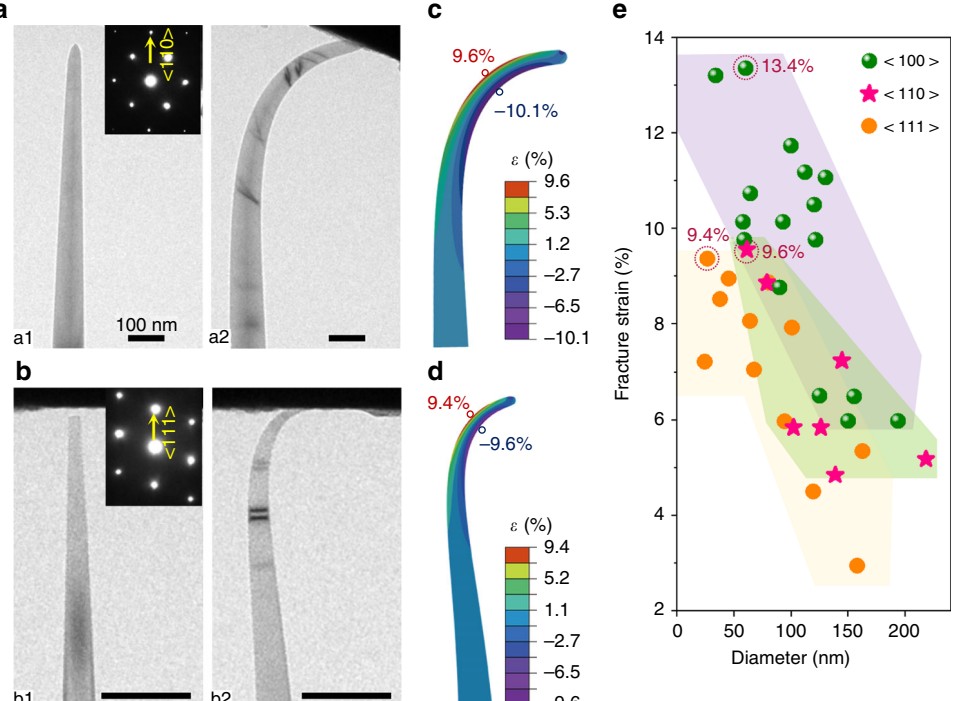

**Fig. 4** Orientation- and size-dependent fracture of diamond nanoneedles. **a, b** TEM images showing the original shape and the maximum deformation immediately before fracture of a <110> (**a**) and a <111> (**b**) nanoneedle. Scalebar is 100 nm. SEAD patterns (insets) indicate orientations in the needles before the tests. **c, d** FEM simulation reproducing critical geometry of the nanoneedles in **a** and **b**, respectively, with maximum principle strain distribution in the nanoneedles. **e** Relationship between size and fracture strain of the <100>, <110>, and <111> nanoneedles.

fractures are {111} cleavage planes (Supplementary Figs. 2–4). Supplementary Figure 5a-c shows the change in electron distribution due to large uniaxial tensile strains along [001]. Charge densities between two adjacent C atoms decrease with increasing strain. At 38% strain, the C–C bond length reaches 1.75 Å, and the charge densities between adjacent C atoms are reduced to such low levels that bonding no longer exists. Calculations for [1$\bar{1}$0] tension show similar results (Supplementary Fig. 6d, e). These results suggest that a critical C–C bond length of 1.75 Å can be used as an indicator for predicting mechanical failure in diamond at atomic scale.

As internal defects are eliminated by reducing the diameter to the sub-100 nm scale, free surface begins to dominate fracture behaviors of diamond nanoneedles. Although great care was taken to prepare high-quality diamond nanoneedles, their surface contained atomic steps of about 1~3 atom in height (Fig.1b). According to Griffith's theory of linear fracture mechanics[3], strength can be greatly reduced by the presence of surface defects. We have simulated surface effects by first principles at nanometer scales, as shown in Fig. 5c-e, where a uniaxial tension is applied along [100], with atomic configurations containing either an atomically flat free surface (Fig. 5c) or free surfaces with atomic steps (Fig. 5d, e). The most elongated C–C bonds tend to be located at the second atomic layer from the surface and neighbored to the surface steps (Fig. 5c-e). These bond lengths increase monotonically with uniaxial tensile strain (Supplementary Fig. 7d). For the atomically flat surface, the critical bond length is reached at a tensile strain of ~28 % (Supplementary Fig. 7a), which is still lower than the ideal strain of 38 % for bulk diamond crystals, due to the presence of free surface. For surfaces with one- and two-atom steps, the tensile strains corresponding to the critical bond length are lowered significantly to 18 and 13.5% (Supplementary Fig. 7b, c), respectively. Further loading will lead to bond breaking and cause an avalanching fracture. In contrast,

the diamond with a two-atom step surface can withstand a critical compressive strain of −20% when loading along the [100] direction (Supplementary Fig. 8). Based on these simulation results, our experimentally measured maximum tensile strain of 13.4% has reached the maximal strain sustainable by free surfaces with two-atom steps for the 60-nm diameter nanoneedles. This is consistent with surface characterization (Fig. 1). It is reasonable to believe that fracture strain of diamond can be further improved in practice by reducing needle diameters and surface defects, as well as total defect density.

Diamond has great potentials in developing high-frequency electronic devices due to its high carrier mobility. However, diamond has an ultra-wide bandgap and is difficult to be doped for bandgap tuning, thereby limiting its applications as a high-performance semiconductor. The ultrahigh elastic strain (13.4%) and fracture strength (125 GPa) of diamond nanoneedles allow great amounts of strain energy to be injected into diamond crystals without inelastic relaxation, thereby increasing the ability to fine-tune physical and chemical properties of diamond[29], by deep elastic strain engineering[30,31].

Our results indicate that achievable fracture tensile elastic strain and strength of diamond nanoneedles depend strongly on diameter, orientation and surface state. We show that by optimizing these properties, ultralarge elastic strain and ultrahigh strength can be achieved in diamond at nanoscales, allowing fine-tuning the physical and chemical properties. The same strategy may be applied to other brittle covalent solids for deep elastic strain engineering.

## Methods

**Fabrication of diamond nanoneedles.** We used natural type Ia diamond to fabricate nanoneedles by FIB milling with a current of 0.5 nA under a voltage of 30 kV. Residual amorphous carbon on as-FIB-milled crystalline surface was then removed by Argon plasma thinning by gradually decreasing voltages from 2.0 kV to 1.5, 1.0, and finally 0.8 kV.

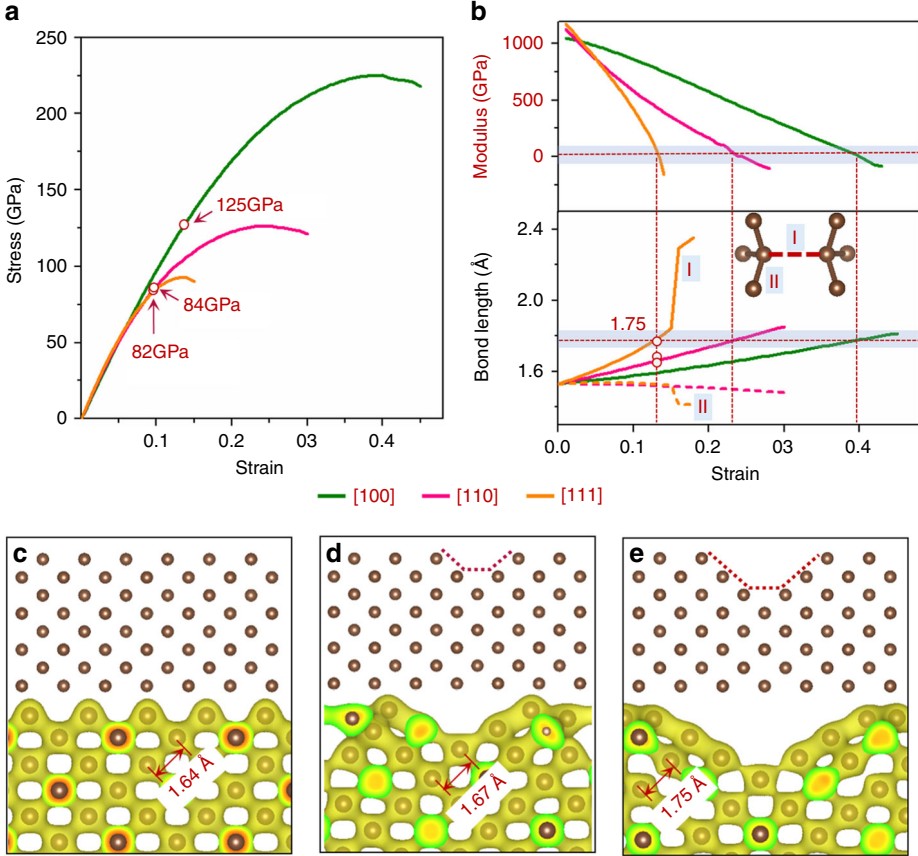

**Fig. 5** First principles simulations of diamond under uniaxial tension with and without a free surface. **a** Theoretical stress-strain relation of uniaxial tension along [100], [110], and [111] directions for a bulk diamond. Experimentally achieved maximum tensile strains are denoted on the curves. **b** Young's moduli and the C–C bond length as a function of uniaxial tensile strain along the [100], [110], and [111] directions. Inset b is a perspective section of the diamond crystal. For tension along [111], the bonds parallel to [111] (denoted as the type-I bond) are elongated, while others (type-II bond) are slightly contracted. For tension along [110], type-I bonds are inclined from the (110) plane while type-II bonds are in the (110) plane. For tension along [100], all bonds belong to the type-I category. The bond lengths discussed in the main text belong to type-I bond. **c–e** Valence charge density at 0.165 e/Bohr3 of a diamond with different surface structures under tension along [100] at strain of 13.4%: **c** atomically flat surface; **d** free surface with a 1-atom step; and **e** free surface with a 2-atom step. The atomic configurations (insets) are constructed based on Fig. 1b. The maximum bond lengths immediately prior to fracture, as denoted in the charge density plots, are plotted as three open circles in the bottom panel of **b**. Clearly, crystals with larger surface defects tend to break first.

**TEM characterization and in situ bending tests**. Characterization of the prepared diamond nanoneedles was carried out using an aberration-corrected FEI Themis Z scanning transmission electron microscope operating at 300 kV. In situ bending experiments were then carried out in a JEM 2100 microscope equipped with a XNano in situ TEM holder (Supplementary Fig. 9a) developed at Center for X-Mechanics, Zhejiang University. The sample was placed on a built-in four-degree freedom (three-dimensional positioning plus self-rotation) nano-manipulator (Supplementary Fig. 9b, c). All three linear motions were precisely driven by built-in piezo actuators with a positioning accuracy of ~0.1 nm and a motion range ~1 mm. A diamond indenter was mounted on a stationary stage, facing the sample holder. By pushing the diamond nanoneedle against the indenter in a step-by-step displacement/loading process, nanoneedles were tested in the TEM. Bending/buckling occurred when the applied load was near Euler's critical point.

**First principles simulations for ideal stress-strain curves of diamond**. We used the Vienna *ab initio* simulation package based on density functional theory within the plane-wave pseudopotential approach[32]. A series of appropriate cuboid unit cells were relaxed to reduce the undesired stress components to 0.02 GPa. The cutoff energy for the plane waves was 800 eV for simulations of a bulk diamond crystal and 350 eV for those with a free surface. The corresponding Monkhorst-Pack K-point meshes used in the calculation were $10 \times 10 \times 10$ and $3 \times 3 \times 3$, respectively.

**FEM analyses**. We used ABAQUS software package (Dassault Systèmes Simulia Corp.)[33] to conduct FEM analyses on models, which replicated the 3D geometry of the nanoneedles. In each analysis the diamond indenter was modeled as a circular plate either inclined or perpendicular to the nanoneedle. Both the nanoneedle and the indenter were treated as deformable solids with the same material properties. A

sliding contact was specified between the tip of the nanoneedle and the top surface of the indenter. Maximum tensile strain of the nanoneedles at the latest frame before fracture was taken as the fracture strain. Contact conditions were adjusted until the shape of the model matched that of diamond nanoneedle in the experiment. Geometric nonlinearity induced by large deformation was taken into account. The small-strain Young's modulus is 1100 GPa and the Poisson's ratio 0.07[25]. To account for the nonlinear elasticity, we use Neo-Hookean nonlinear elasticity model of the form of $U = C_{10}(\overline{I_1} - 3) + \frac{1}{D_1}(J_{el} - 1)^2$, where, $U$, $\overline{I_1}$ and $J_{el}$ represents the strain energy, the first strain invariants and the elastic volume strain, respectively. $C_{10}$ and $D_1$ are material constants of 257 GPa and $5.45 \times 10^{-3}$ GPa$^{-1}$, respectively[25]. Friction during bending tests may have an impact on the compression force and consequently affect stress distribution within the nanoneedle. Here, we estimate friction effects through Euler instability of a slender pillar (the diamond nanoneedle). To the first-order approximation by neglecting crystal anisotropy, the longitudinal compression stress on the needle is given by Euler's critical stress $\sigma_{cr}$, according to Eq. (1):[34]

$$\sigma_{cr} = \frac{\pi^2 E}{(KL/r)^2} \qquad (1)$$

where $E$ is the isotropic elastic modulus of diamond, $K$ is effective length factor, $L$ is the unsupported length of column, $r$ is the radius of gyration. Given $E = 1050$ GPa, $K = 0.7$, $L = 4000$ nm and $r = 50$ nm, the resultant $\sigma_{cr}$ is ~3 GPa. Compared to the strength on the order of 100 GPa measured in diamond, the friction effect appears negligible and only contributes to uncertainties.

## Data availability

The data that support the findings of this study are available from the corresponding author upon reasonable request.

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

## Acknowledgements

This work was supported by the National Key R&D Program of China (2018YFA0703400), and the Natural Science Foundation of China (11725210, 11702165, and 11672355). We acknowledge helpful discussions with Prof. Sulin Zhang at Pennsylvania State University.

## Author contributions

Y.J.T. and W.Y. initiated the project and created the experimental protocols. H.T.W., Y.Z.Z., and J.B.L developed XNano in situ TEM holder. P.H.L, T.Y.J., and Z.S. carried out the fabrication of diamond nanoneedles under the direction of A.M.N. Y.Q.B., and A.M.N conducted the in situ TEM testing. Z.Y.L., J.L.H., and Y.B.W. analyses the data. Y.Q.B. and H.T.W. performed FEM and DFT calculations. A.N., H.T.W., Y.J.T., and W.Y. wrote the manuscript and all the authors contributed to the discussion and revision of the manuscript.

## Competing interests

The authors declare no competing interests.
