## [Peer Review File · Nature Communications]

Reviewers' comments:

Reviewer #1 (Remarks to the Author):

This paper is a follow-up of reference [3] in the manuscript. The same experimental techniques are used to manufacture the diamond nano needles, to test the mechanical behaviour and to determine the maximum strain (FEM and DFT). The only novelty of the paper is that needles with different orientations ($\langle 100 \rangle$, $\langle 110 \rangle$ and $\langle 111 \rangle$) have been analysed and in all cases the ideal strength has been achieved because of the lack of defects in the nano needles. Moreover, it was found that fracture always took place along the $\{111\}$ facets of the diamond crystals.

All these results are interesting and the paper is well written, I think that advance in knowledge provided by the paper is very limited (in comparison with reference [3]) and does not justify its publication in Nature Communications. I recommend that the authors submit the paper to other journal related to nanotechnology or extreme mechanics

Reviewer #2 (Remarks to the Author):

In this work the authors report on an experimental study, supported by computational simulations and theoretical analysis, of large tensile strain and stress achieved in diamond nanoneedles that approach theoretical limits. This phenomenon was recently reported by another group, but the present findings expand considerably the range and depth of the observed behaviors, especially the crystal size and orientation dependence of the measured mechanical properties. The reported results are interesting and insightful, providing a useful foundation for further exploration and application of this remarkable phenomenon. The experimental measurements seem to have been done carefully and presented well. There are, however, some issues with the analysis and discussion, which are listed below, that need to be addressed before a decision regarding a final recommendation for this manuscript can be reached.

(1) In describing the bent diamond nanoneedles, the imaging and illustrations throughout the manuscript depict the tensile strains on the convex (outer) side of the nanoneedles while the concave (inner) side is labeled as having zero strains. This cannot be the case since there must be compressive, i.e., negative, strains on the concave side, which need to be properly considered and described.

(2) Given the existence of compressive strains on the concave side of the nanoneedles, a natural follow-up question is then where does the fracture initiate? Does it always occur on the convex side, or the concave side, or is it size and orientation dependent?

(3) There is obviously a large strain gradient in the bent nanoneedle. How would such a gradient affect the fracture mechanism and breaking strain and peak stress?

(4) There are some descriptions and discussions in the manuscript that need correction, further consideration and/or clarification.

4.1 - line 181, "... progressively smaller strains are required ..." should read "... progressively larger strains are required ..."

4.2 - What is the meaning of the three open circles in Fig. 5b?

4.3 - line 396-397, the statement "For tension along $[100]$, all bonds belong to the type-I category" is incorrect.

4.4 - In Fig. S5, the $[-110]$ and $[100]$ crystal directions are depicted as if they were perpendicular

to each other, but they actually are not.

4.5 - The Griffith theoretical strength given by $E/10$ or $E/9$ is non-directional, but this quantity is being compared to the measured maximal strength in the [100] direction.

4.6 - Line 214-216, discussion on possible further increasing fracture strain by reducing needle size and surface defects, but this would lead to maximal strength exceeding the Griffith limit since the currently measured tensile strength along the [100] direction has already reached the theoretical limit.

4.7 - The reported maximal strain on [100] needle is 13.4%, which is close to the theoretical strain of 13.5% calculated for the needle with 2-atom steps, but the stress was derived from bulk diamond stress-strain relation at 13.4% to reach the conclusion that the extracted, again from the bulk diamond stress-strain relation, stress value of 125 GPa at 13.4% matches the Griffith limit of 122 GPa. These comparisons are inconsistent.

4.8 - Line 182-192, are authors talking about the same [111] bond length under different tensile stress here? Please clarify.

Reviewer #3 (Remarks to the Author):

The authors performed in-situ TEM nanomechanical experiments on diamond nanoneedles and systematically explored the size- and orientation-dependence of elastic strain and strength of nanoscale diamond. The $\langle 100 \rangle$ -oriented nanoneedles with a diameter of 60 nm was found to have the highest elastic tensile strain of 13.4% and the corresponding tensile strength was estimated to be 125 GPa. The reported values are comparable with the theoretical elasticity and Griffith strength limits of diamond. Careful first principles modeling results suggest that maximum achievable elastic strain and the corresponding strength are closely correlated to surface defects of the diamond nanoneedles. The authors' main findings are indeed convincing and robust, however there are a few additional suggestions and clarifications that the authors should consider.

1) The authors adopted the elastic parameters used in ref.(3) for FEM simulations to estimate the maximum strength (stress) corresponding to the in-situ TEM observations. However, the elastic parameters used in ref.(3) were along [111] direction. As we know single crystalline diamond has a finite elasticity anisotropy along different directions, the FEM simulations along [100] and [110] should consider the influence due to elasticity anisotropy, so the estimated maximum strength values would be more accurate.

2) Although the assumed friction probably won't significantly change the results, the description on the friction in estimating the maximum strength was not clear. The choice of friction may have an impact on the compression force experienced by the nanoneedle and consequently affect the stress distribution within the nanoneedle. Some discussions should be provided.

3) Lines 165-168: It is not clear how the projection is done from [100] or [110] to [111].

4) Lines 249-250: The statement that "Bending/buckling occurred when the applied load reached Euler's critical point" might not be accurate here, since the experiments were not done with ideal conditions (i.e. perfect alignment and geometry, etc).

5) Line 159: the reference should be (3) instead of (2).

Point-by-Point Responses to the Referees

Report of Reviewer #1 (Remarks to the Author):

This paper is a follow-up of reference [3] in the manuscript. The same experimental techniques are used to manufacture the diamond nano needles, to test the mechanical behaviour and to determine the maximum strain (FEM and DFT). The only novelty of the paper is that needles with different orientations ($\langle 100 \rangle$, $\langle 110 \rangle$ and $\langle 111 \rangle$) have been analysed and in all cases the ideal strength has been achieved because of the lack of defects in the nanoneedles. Moreover, it was found that fracture always took place along the $\{111\}$ facets of the diamond crystals.

All these results are interesting and the paper is well written, I think that advance in knowledge provided by the paper is very limited (in comparison with reference [3]) and does not justify its publication in Nature Communications. I recommend that the authors submit the paper to other journal related to nanotechnology or extreme mechanics

Response: We thank the referee for the critical comments. However, we respectfully disagree with the comments “that advance in knowledge provided by the paper is very limited (in comparison with reference [3])”.

Please allow us to summarize the major differences between our study and Reference 3 (Science 360, 300-302, 2018). (1) Different fabrication method. The diamond nanoneedles were prepared by bias-assisted reactive ion etching in Reference 3, but by utilizing focused ion beam (FIB) milling coupled with argon plasma thinning in our study. Our method allowed precise control on the size, orientation and surface conditions of the diamond nanoneedles. This new fabrication method ensured us to obtain, for the first time, vital information on how strains developed in the diamond nanoneedles with various diameters and orientations. This contrasts the results of Reference 3, in which only one type of diamond nanoneedle with $\langle 111 \rangle$ orientation with one diameter of ~ 300 nm was studied. (2) Direct evaluation on strain distribution in nanoneedles. In contrast to the *in situ* SEM method employed in Reference 3, the *in situ* TEM measurement used in our work provides more solid experimental evidence of strain distribution by directly capturing distortion of the diamond lattice through supplementary electron diffraction with high spatial-temporal resolution. No such information was obtained in Reference 3. (3) Significantly improved mechanical properties of nanoneedles. We report the maximum achievable tensile strain (13.4%)

and strength (125 GPa) of diamond nanoneedles. These values are approximately 50% and 30% respectively greater than those reported in Reference 3, and are the largest values that have ever been experimentally achieved in diamond so far. Thus we have demonstrated for the first time that experimental strength of diamond nanoneedles can indeed reach the Griffith theoretical strength (~122 GPa). (4) Most importantly, our experimental data, together with first principles simulations, demonstrate that elastic deformation of diamond nanoneedles highly depends on their diameter, orientation, and surface conditions. Besides diameter reduction and orientation selection, smoothness of the surface is critical to make diamond nanoneedle the strongest three-dimensional material. Our work demonstrates that, by optimizing diameter, orientation, and surface conditions, ultra-large elastic strain and ultrahigh strength can be achieved, allowing fine-tuning physical and chemical properties.

We would also like to draw an attention to the significance of technical development reported in our work. As a matter of fact, our project was initiated several years ago. However, it was challenging to perform the desired mechanical tests on diamond using commercial *in-situ* TEM holders inside the TEM, because the mN-level forces needed to break a nanoscale diamond require high-precision displacement control. We spent more than two years to develop our own *in situ* mechanical testing system inside the TEM (X-Nano *in-situ* TEM instrument, Fig. S9). Thanks to this X-Nano *in-situ* TEM instrument, significant progress has been made, enabling us to gain more comprehensive insight into the strength and deformation limits of diamond.

Report of Reviewer #2 (Remarks to the Author):

In this work the authors report on an experimental study, supported by computational simulations and theoretical analysis, of large tensile strain and stress achieved in diamond nanoneedles that approach theoretical limits. This phenomenon was recently reported by another group, but the present findings expand considerably the range and depth of the observed behaviors, especially the crystal size and orientation dependence of the measured mechanical properties. The reported results are interesting and insightful, providing a useful foundation for further exploration and application of this remarkable phenomenon. The experimental measurements seem to have been done carefully and presented well. There are, however, some issues with the analysis and discussion, which are listed below, that need to be addressed before a decision regarding a final recommendation for this manuscript can be reached.

(1) In describing the bent diamond nanoneedles, the imaging and illustrations throughout the manuscript depict the tensile strains on the convex (outer) side of the nanoneedles while the concave (inner) side is labeled as having zero strains. This cannot be the case since there must be compressive, i.e., negative, strains on the concave side, which need to be properly considered and described.

Response: We thank the reviewer for the constructive comments. Following the reviewer's suggestion, we have included the maximum compressive strain on the concave side of the diamond nanoneedles in the revised manuscript (Figure 2c, Figure 3b, Figure 4c, Figure 4d, Figures S3b, and Figure S4b). Corresponding discussion has also been included in the revised manuscript.

“Besides the tensile strain on the convex side of the nanoneedles, there must be a compressive strain on the concave side for mechanical equilibrium. FEM calculations show that the absolute values of the compressive strain on the concave side are slightly larger than those of the corresponding tensile strain at the convex side (Fig. 2c, Fig.3b, Fig.4c, Fig.4d, Fig. S3b and Fig. S4b). The tensile strength of a material is usually much lower than the compressive strength because compressive loading tends to suppress microcracks. It has been experimentally demonstrated that compressive strains of -19% and -16% can be achieved in the <100> and <111>-oriented diamond pillars with a diameter of 200 nm in uniaxial compression, respectively¹⁹. Here, the compressive strains at the concave side of diamond nanoneedles are still far below the limits of the nanoneedles. Therefore, fracture tends to initiate from the convex side and the tensile limits of the nanoneedles can be well expressed by the bending experiments³.” (Line 134-146, Page 7)

(2) Given the existence of compressive strains on the concave side of the nanoneedles, a natural follow-up question is then where does the fracture initiate? Does it always occur on the convex side, or the concave side, or is it size and orientation dependent?

Response: Thanks for this insight. Although the absolute values of the compressive strain on the concave side are slightly larger than those of the corresponding tensile strain at the convex side of diamond nanoneedles, we believe the fracture should initiate on the surface of the convex side with maximum tensile strain. The main reasons are as follows:

(1) In practice, the compressive deformation limit of diamond is higher than the tensile one because compressive loading tends to suppress microcracks. It has been experimentally demonstrated that -19% and -16% compressive strains can be respectively achieved in the <100> and <111>-oriented diamond pillars with a

diameter of 200 nm in uniaxial compression (Nano Letters, 16, 812-816, 2016). In this work, compressive strains of $\langle 100 \rangle$ and $\langle 111 \rangle$ -oriented diamond nanoneedles only approached -14% and -9.6%, respectively, which are believed still far below their limits of the nanoneedles. In contrast, the tensile strain at the convex side, which has a parallel value to the compressive one at the concave side, almost approached the tensile deformation limit of diamond.

(2) The first-principle calculations also verify the diamond can withstand a larger compressive strain than tensile one before failure. We performed the first-principle calculations on the compressive deformation of diamond with consideration of a 2-atom surface flaw. The initial model is the same one used for tensile deformation in Figure S7c. As shown in Fig. R1, the diamond supercell was compressed along the $\langle 100 \rangle$ direction. The bond around the surface flaw highlighted by the red arrow was stretched with the increase of compressive strain. Figure R1a-d show the charge density distribution of the diamond with different compressive strains. We can see that the charge is depleted at a compressive strain of -20%, indicating the breakage of the bond near the surface. As shown in Figure R1e, when the compressive strain approached 20%, the bond length reached 1.75\AA , which was believed as a critical value indicating the bond breaking and fracture initiation. For the diamond crystal with 2-atom surface flaws, its compressive strain limit could approach 20%, which is significantly higher than the tensile one of 13.5%.

Therefore, fracture tends to initiate on the surface of the convex side with maximum tensile strain and the tensile limits of the nanoneedles can be well expressed by the bending experiments. Corresponding simulations have been included in the supplementary as Fig S8.

Figure S8. **The first principle simulations of a diamond under [100] uniaxial compression with two-step surface flaw.** a-c, Surface plots of the valence charge density at 0.165 e/Bohr^3 under [100]-direction compressive strains of $\epsilon = -3\%$; $\epsilon = -9\%$; $\epsilon = -15\%$ and $\epsilon = -20\%$, respectively. e. The maximal bond lengths denoted in the charge density plot change with the uniaxial tensile strain.

Corresponding discussion has also been included in the revised manuscript.

“In contrast, the diamond with a two-atom step surface can withstand a critical compressive strain of -20% when loading along the [100] direction (Supplementary Fig. S8).” (Line 224-226, Page 11)

(3) *There is obviously a large strain gradient in the bent nanoneedle. How would such a gradient affect the fracture mechanism and breaking strain and peak stress?*

Response: The strain gradient is usually generated in the bending or buckling pillars. In our work, there do exist obvious strain gradient in the bent diamond nanoneedles before their fractures. However, differing from the materials with good ductility (plasticity) like metals, the diamond nanoneedles exhibited a sudden brittle fracture, giving no time for the observation of the strain gradient's role in the fracture or cracking of the diamond at present temporal resolution inside TEM. In spite of various strain gradient existing among different diamond nanoneedles, the diamond nanoneedles exhibit common feature of brittle fracture of {111} cleavage. Since the strain gradient is complicatedly interwoven with the size, curvature, and stress state of the sample, it is a challenge to get a direct answer to the correlations between the strain gradient and breaking strain as well as the peak stress. However, the strain gradient is no doubt a subject worthy of our future study, requiring a rational design of series of experiments.

(4) *There are some descriptions and discussions in the manuscript that need correction, further consideration and/or clarification.*

4.1 - line 181, "... progressively smaller strains are required ..." should read "... progressively larger strains are required ..."

Response: Sorry for the mistake. Corresponding correction has been made in the revised manuscript.

“For diamond stretched along [110] and [100], progressively larger strains are required to reach zero Young's moduli (top panel of Fig. 5b)” (Line 193-195, Page 10)

4.2 - What is the meaning of the three open circles in Fig. 5b?

Response: Three open circles from bottom to the top of Fig. 5b are corresponding to the lengths of C-C bond labeled in Fig 5c-e, respectively. We have added a sentence in the caption of Fig.5 to explain the meaning of the three open circles in Fig. 5b. See the below:

“The maximum bond lengths immediately prior to fracture, as denoted in the charge density plots, are plotted as three open circles in the bottom panel of b.” (Line 427-429, Page 21)

4.3 - line 396-397, the statement "For tension along [100], all bonds belong to the type-I category" is incorrect.

Response: Such a statement was made according to the results of our first-principles calculations, which indicate that all bonds are elongated when stretching the diamond supercell along the [100] direction. This can also be understood from the perspective of geometric relationships. As shown in Fig. R1 below, A, B, C and D represent four chemical bonds of a given C atom, while A', B', C' and D' are their corresponding equivalent bonds in the supercell, under tension along [100]. As indicated by red arrows in Fig. R1, bond A, B', C and D all receive a resolved tensile stress along the bond orientation. Therefore, all bonds are elongated and thus belong to type-I category. Similar descriptions can also be found in other literature (*Computer Physics Communications* 238, 244-253, 2019; *Physical Review Letters* 84, 5160-5163, 2000).

Figure R1. The atomic model indicating an equivalent orientation of all the bonds relative to the [100] direction.

4.4 - In Fig. S5, the [-110] and [100] crystal directions are depicted as if they were

perpendicular to each other, but they actually are not.

Response: Sorry for the typo. The projected direction should be $[110]$ and two perpendicular directions should be $[1-10]$ and $[001]$, respectively. See below:

Fig S6

4.5 - The Griffith theoretical strength given by $E/10$ or $E/9$ is non-directional, but this quantity is being compared to the measured maximal strength in the $[100]$ direction.

Response: Griffith theoretical strength is derived from a continuum model for the crack extension, assuming the isotropy and homogeneity of the substance, as well as the linearity (Phil. Trans. Roy. Soc. Lond. A 221, 163-198, 1921). It is a well-known theoretical strength for brittle materials, and regarded as a rough “benchmark” to judge whether the material approaches in a theoretical strength scope. Although anisotropy and specific crystal structures were not considered in his model, Griffith agreed that the best arrangement for withstanding tension is where the strongest directions of all the “molecules” are parallel to the axis of tension (Phil. Trans. Roy. Soc. Lond. A 221, 163-198, 1921). In our work, the maximal strengths of $[100]$, $[110]$, and $[111]$ -orientated diamond nanoneedles are 125 GPa, 84 GPa and 82 GPa, respectively. Therefore, we compared the maximal strength among all the nanoneedles with the theoretical strength limit of $E/9$, aiming to provide a strategy for approaching the highest strength of this solid. Of course, we can also compare this value with those of the highest strength experimentally achieved in other well-known

strong materials like carbon nanotube (~110 GPa, Nature nanotechnology 3, 626, 2008) and graphene (120 GPa, Science 340, 1073-1076, 2013), demonstrating this outstanding strength of the diamond nanoneedles.

4.6 - Line 214-216, discussion on possible further increasing fracture strain by reducing needle size and surface defects, but this would lead to maximal strength exceeding the Griffith limit since the currently measured tensile strength along the [100] direction has already reached the theoretical limit.

Response: Thanks again for the comments. As we mentioned in the above response, the Griffith limit of $E/9$ is a roughly estimated value by approximation. In addition, numerous attempts were made to determine the theoretical strength more accurately. However, there are rather large differences in its estimates depending on the approximations and calculation procedures. For example, Slutsker obtained values of theoretical strength for 15 metals fall in the range $E/6$ - $E/10$ (Fiz Tverd Tela 46, 1606-1613, 2004). Mecholsky used fractal geometry to estimate the theoretical strength of brittle materials approximately $E/8$ (Materials Letters 60, 2485-2488, 2006). We can see that the Griffith limit is not strictly an absolute criterion for the upper limits on the theoretical tensile strength. The comparison between our experimental strength and Griffith theoretical strength is just one of the ways to show how far the strength of the diamond can go. Correspondingly, first principle calculations, which is widely recognized as a more accurate way to predict the ideal strength of diamond, is also present in our work. The ideal strength of <100>-orientated diamond calculated by the first principle calculations is as high as 225 GPa (Figure 5a), which is much higher than our experimental value. When considering the effect of surface (flat surface, 1-atom step, and 2-atom step), the calculated strength of diamond is still as high as 205 GPa (28% strain), 157 GPa (18% strain), and 125.4 GPa (13.5% strain) (Figure S7), respectively, which are still greater than the experimentally achieved value. Our first principle calculations indicate that higher strength than the Griffith theoretical limit could be achieved if the defect-free diamond nanoneedles are with a perfectly smooth surface, offering guidance for the ultra-strong materials design.

4.7 - The reported maximal strain on [100] needle is 13.4%, which is close to the theoretical strain of 13.5% calculated for the needle with 2-atom steps, but the stress was derived from bulk diamond stress-strain relation at 13.4% to reach the conclusion that the extracted, again from the bulk diamond stress-strain relation, stress value of 125 GPa at 13.4% matches the Griffith limit of 122 GPa. These comparisons are inconsistent.

Response: Thanks for the insight. I think it has been explained well for this comment in the response for comments 4.5 and 4.7. The Griffith limit of $E/9$ was an estimated value by an approximation based on energy consideration. It stands to reason the maximum value (125 GPa) of our experimental data goes little beyond the value of $E/9$ (122 GPa), as $E/9$ here is usually roughly considered as “benchmark” to judge whether the material approaches in an ideal strength scope. As for the first principle calculations, when considering a 2-atom-step flaw on the surface, the calculated elastic strain limits and strength are about 13.5% and 125.4 GPa, respectively, which are also closely approached by the highest values obtained in our experiments. The Griffith theory and the first principle calculations are just two different calculation procedures for setting the upper limits for the theoretical tensile strength of diamond. Despite of a tiny discrepancy between the theoretically calculated values, all the results indicate that our experimental value has approached to the scope of the ideal strength. We believe there is no conflict in our conclusions, since we get the straightforward experimental values for the strain and strength.

4.8 - Line 182-192, are authors talking about the same $[111]$ bond length under different tensile stress here? Please clarify.

Response: Thank the reviewer for the kindly concern. The relationships between fracture and bond length that we talked about are all about the type-I $\langle 111 \rangle$ bonds that are elongated. We added one word in the caption of Figure 5 for the clarification: “The bond lengths discussed in the main text belong to type-I bond.” (Line 423-424, Page 21)

Report of Reviewer #3 (Remarks to the Author):

The authors performed in-situ TEM nanomechanical experiments on diamond nanoneedles and systematically explored the size- and orientation-dependence of elastic strain and strength of nanoscale diamond. The $\langle 100 \rangle$ -oriented nanoneedles with a diameter of 60 nm was found to have the highest elastic tensile strain of 13.4% and the corresponding tensile strength was estimated to be 125 GPa. The reported values are comparable with the theoretical elasticity and Griffith strength limits of diamond. Careful first principles modeling results suggest that maximum achievable elastic strain and the corresponding strength are closely correlated to surface defects of the diamond nanoneedles. The authors' main findings are indeed convincing and robust, however there are a few additional suggestions and clarifications that the authors should consider.

1) The authors adopted the elastic parameters used in ref.(3) for FEM simulations to estimate the maximum strength (stress) corresponding to the in-situ TEM observations. However, the elastic parameters used in ref.(3) were along [111] direction. As we know single crystalline diamond has a finite elasticity anisotropy along different directions, the FEM simulations along [100] and [110] should consider the influence due to elasticity anisotropy, so the estimated maximum strength values would be more accurate.

Response: We agree with the referee's comment that "the FEM simulations along [100] and [110] should consider the influence due to elasticity anisotropy, so the estimated maximum strength values would be more accurate". We should also pay attention to the influence of modulus nonlinearity on the accuracy of strength calculations. In this study, therefore, we estimate the strength by introducing the observed strain values and calculated the stress by using the first principle calculations instead of FEM simulations, including the consideration of orientation, elasticity anisotropy, and nonlinearity of the diamond. In this way, the calculated strength corresponding to the same strain is actually lower than that estimated by the FEM method used in Reference 3, especially in large strain regions. For example, the calculated strength values corresponding to the strain of 13.4% are 125 GPa by first principle calculations (this work) and 147.4 GPa by FEM simulations (Reference 3), respectively.

2) *Although the assumed friction probably won't significantly change the results, the description on the friction in estimating the maximum strength was not clear. The choice of friction may have an impact on the compression force experienced by the nanoneedle and consequently affect the stress distribution within the nanoneedle. Some discussions should be provided.*

Response: We thank the reviewer for the insightful comment and suggestion. To clarify the friction effect, we first investigate how much compression stress can drive the buckling instability of the diamond nanoneedles. On first-order approximation, the longitudinal compression stress on a nanoneedle can be estimated by Euler's critical stress, according to the following formula (Theory of Elasticity, Pergamon press 1981):

$$\sigma_{cr} = \frac{\pi^2 E}{(KL/r)^2} \quad (1)$$

Where σ_{cr} is the Euler's critical stress, E is the elastic modulus of diamond, K is the effective length factor, L is the unsupported length of the needle, and r is the radius of

gyration. Given $E = 1050$ GPa, $K = 0.7$, $L = 4000$ nm and $r = 50$ nm, the resultant σ_{cr} is ~ 3 GPa, without considering anisotropy of diamond. It means that only a compression stress on the order of 3 GPa can reach Euler buckling instability of the diamond nanoneedles. Compared to a strength of 125 GPa measured in the diamond, the value of σ_{cr} can be included in the tolerance range in our work. Therefore, the friction effect can be negligible, as it only affects a negligible compressive stress. The corresponding discussion has been included in the revised manuscript:

“Friction during bending tests may have an impact on the compression force and consequently affect stress distribution within the nanoneedle. Here, we estimate friction effects through Euler instability of a slender pillar (the diamond nanoneedle). To the first-order approximation by neglecting crystal anisotropy, the longitudinal compression stress on the needle is given by Euler's critical stress σ_{cr} , according to the following formula³⁵:

$$\sigma_{cr} = \frac{\pi^2 E}{(KL/r)^2} \quad (1)$$

where E is the isotropic elastic modulus of diamond, K is effective length factor, L is the unsupported length of column, r is the radius of gyration. Given $E = 1050$ GPa, $K = 0.7$, $L = 4000$ nm and $r = 50$ nm, the resultant σ_{cr} is ~ 3 GPa. Compared to the strength on the order of 100 GPa measured in diamond, the friction effect appears negligible and only contributes to uncertainties.” (Line 287-299, Page 14 and 15)

3) Lines 165-168: *It is not clear how the projection is done from [100] or [110] to [111].*

Response: The projection is meant to resolve the stress onto the $\langle 111 \rangle$ -orientated C-C bonds. Figure R2 shows the detailed stress decomposition process for the $\langle 100 \rangle$ -loading case, where the lattice of diamond is elongated into a tetragonal configuration at the largest strain of 38%. The right panel of **a** is a sectional view corresponding to the plane outlined by the red lines in the left panel. The (111) plane is viewed along edge-on in the right panel. The normal stress on the (111) plane ($\sigma_{[111]}$) is calculated as the following:

$$\sigma_{[111]} = \sigma_{[001]} \cos^2 \alpha,$$

$$\cos \alpha = \frac{1 + \varepsilon}{\sqrt{1 + 1 + (1 + \varepsilon)^2}}$$

where ε is the strain along [001] direction, α is the angle between $[1, 1, 1 + \varepsilon]$ and $[0, 0, 1 + \varepsilon]$ directions in the severely deformed lattice. The ideal strength approached in the $\langle 100 \rangle$ -orientated diamond nanoneedles is $\sigma_{[001]} = 225$ GPa, when the $\varepsilon = 38\%$. The resolved stress along the C-C bonds in the [111] direction is $\sigma_{[111]} = 109$ GPa.

For the $\langle 110 \rangle$ -loading case, the projection process is similar to that of $\langle 100 \rangle$ -loading case, as shown in Fig. R2b. Here, $\sigma_{[111]} = \sigma_{[001]} \cos^2 \alpha$, $\cos \alpha = \frac{\sqrt{2}(1+\varepsilon)}{\sqrt{2(1+\varepsilon)^2+1}}$, where ε is strain along $[110]$, α is the angle between $[1+\varepsilon, 1+\varepsilon, 1]$

and $[1+\varepsilon, 1+\varepsilon, 0]$ in the severe deformed lattice. The ideal strength approached in the $\langle 100 \rangle$ -orientated diamond nanoneedles is $\sigma_{[001]} = 126$ GPa, when the $\varepsilon = 26\%$. the resolved stress along the C-C bond in the $[111]$ direction is $\sigma_{[111]} = 95$ GPa.

Therefore, when the ideal strengths along $[100]$ and $[110]$ are projected onto $[111]$ -orientated C-C bonds, the resolved ideal strengths are values of 109 and 95 GPa, respectively, which are in good agreement with the ideal strength of 92 GPa along $[111]$. The projection illustrates that fracture is controlled by the breakage of $\langle 111 \rangle$ -orientated C-C bonds, explaining the fractures along the $\{111\}$ planes. We have added supplementary Fig. S5 to illustrate these projection processes.

Fig S5 Illustration of the ideal strength projection from (a) $[100]$ and (b) $[110]$ to $[111]$.

4) Lines 249-250: The statement that “Bending/buckling occurred when the applied load reached Euler’s critical point” might not be accurate here, since the experiments were not done with ideal conditions (i.e. perfect alignment and geometry, etc).

Response: We have revised that statement to “Bending/buckling occurred when the applied load was near Euler’s critical point.” (Line 264-265, Page 13)

5) *Line 159: the reference should be (3) instead of (2).*

Response: We have revised it in the current version.

REVIEWERS' COMMENTS:

Reviewer #2 (Remarks to the Author):

The authors have addressed most questions and suggestions raised in my last report, and have made proper revisions in the manuscript. While there are still some remaining issues for further studies, I find the revised manuscript now suitable for publication in Nature Communications.

Reviewer #3 (Remarks to the Author):

The authors carefully addressed the concerns raised by the reviewer reasonably well. The reviewer would recommend its publication.